# A Model to Manage the Lane-Changing Conflict for Automated Vehicles Based on Game Theory

**Liling Zhu** [1], **Da Yang** [2], **Zhiwei Cheng** [2], **Xiaoyue Yu** [2] and **Bin Zheng** [2,*]

1    School of Business, Sichuan Normal University, No. 1819 Chenglonglu, Chengdu 610101, China
2    School of Transportation and Logistics, Southwest Jiaotong University, No. 999 Xi'an Road, Chengdu 611756, China
*    Correspondence: binzheng@swjtu.edu.cn

**Abstract:** In this study, we propose a lane-changing conflict management model based on game theory for automated vehicles. When a vehicle plans to change to the adjacent lane, and if there is a closely following vehicle on that lane, the following vehicle must sacrifice its speed to make space for the lane-changing vehicle, which means there are conflicts of interest between two vehicles. So far, there is no clear answer if the following vehicle should make space for the lane-changing vehicle. These individualistic lane-changing models may lead to suboptimal traffic flow or even traffic safety issues. To solve this problem, this study designed a model based on game theory to solve lane-changing conflicts between the lane-changing vehicle and the following vehicle in the target lane. When the two vehicles enter a lane-changing conflict, the payoffs of the two vehicles under various combinations of strategies were evaluated, and the final strategy and the acceleration for each vehicle were obtained based on the principle of benefit equilibrium. The simulation is conducted to analyze the game strategy of the lane-changing vehicle (LV) and the close rear vehicle (RV) in the process of lane-changing from different initial positions. The results show that, under the hypothesis scenario in the simulation, the strategy {changing a lane, avoiding } will be chosen when the RV is initially located in the range of [0, 40 m], while {not changing a lane, not avoiding} is more appropriate when the initial position of the RV is in the range of [41 m, 90 m].

**Keywords:** transportation; automated vehicles; game theory; lane-changing

## 1. Introduction

In recent years, automated vehicles have received extensive attention and become a hot issue [1–5]. Autonomous driving can solve the problem of urban road congestion and greatly reduce traffic congestion [6–11]. However, the safety of automated vehicles has become the primary issue of autonomous driving research. Any automated vehicle is inseparable from the study of safety [12–17], and lane-changing decisions during driving will have a great impact on the safety of automated vehicles [18–21]. This study focuses on the lane-changing control of automated vehicles, which is one of the essential driving maneuvers of vehicles and has a key impact on driving safety and traffic efficiency.

A lane-changing process of an automated vehicle is shown in Figure 1. In a lane-changing process, four vehicles are potentially involved, including the lane-changing vehicle (LV), the preceding vehicle on the current lane (PV), the front vehicle on the target lane (FV), and the rear vehicle on the target lane (RV). We often see this in our driving. When the LV attempts to change to the target lane, and the longitudinal distance between the LV and RV is not large enough to ensure lane-changing safety, the RV needs to create a safe lane-changing space for the LV by deceleration. The RV sacrifices its speed to benefit the LV, which means the interests of the LV and RV have interest conflicts [22–24]. In this situation, the RV and the LV must cooperate to finish the lane-changing process. This cooperation is easy for human drivers, as they have the inherent ability to manage conflicts [25]. However,

it is difficult for automated vehicles without an elaborate conflict management design to deal with such a situation [26]. Therefore, this paper attempts to design a lane-changing conflict management strategy to help automated vehicles change a lane safely when there is a conflict between the LV and RV.

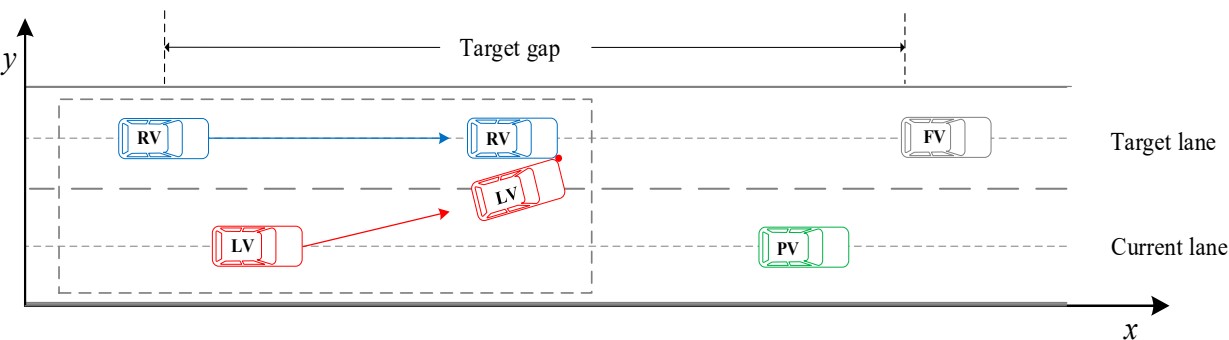

**Figure 1.** Lane-changing conflict scenario.

In the existing studies, there are three methods to address the interest-conflict problem between LV and RV. The first method is mimicking human drivers' lane-changing behavior, in which each vehicle responds and makes a behavior decision according to the instantaneous traffic condition [27]. The second method simply forbids lane-changing when there is a conflict to guarantee safety [28]. The third method is that the RV decelerates to produce a safe space for the LV [29]. All three methods have their own shortcomings. In the first method, each vehicle makes a decision based on the predictions of how the other vehicles will act. If the prediction is wrong, it may cause crashes, so safety cannot be guaranteed [27]. The second method can avoid safety problems; however, it increases the difficulty of lane-changing, which may cause a missed turn of the LV [28]. For example, if a vehicle attempts to arrive at an off-ramp and then leave the freeway, if the second method is applied, lane-changing may be prevented, resulting in missing the off-ramp for the LV. The third method can also ensure safety, while the RV needs to sacrifice its interest for the LV, which is not fair to the RV. If the two vehicles are from different vehicle manufacturers, the agreement may be very difficult to reach [29].

To overcome the shortcomings of the existing models, this paper proposes a conflict management model for the lane-changing of automated vehicles, borrowing the inherent balancing ability of game theory. Since the game theory is capable of obtaining a balance for the two interest-conflicting subjects, and if both vehicles execute the conflict resolution mechanism proposed in this paper, safety and fairness can be guaranteed, whether they belong to the same vehicle manufacturing company or not [30]. The paper first defines the potential conflict points between the lane-changing vehicle and the vehicle behind the target lane and formulates the criteria for the occurrence conditions of the game based on the initial position and velocity of the vehicle. Secondly, considering the benefits of the vehicles involved in the lane-changing process of automatic driving, the kinematics method is used to deduce the corresponding revenue value under different strategy combinations, and then the acceleration selection model was used to obtain the corresponding longitudinal acceleration of the two vehicles under each strategy, and the final strategy of the two vehicles was determine. Finally, automated vehicles encountering conflicts in the process of lane-changing do not need to play games but drive directly according to the results derived from the proposed game theory model in this paper. The proposed model is a general model for the autonomous driving on mainline lanes of a freeway, and the special cases (like merging and weaving) are not discussed in this paper.

The key contributions of this study can be summarized as follows:

- Most of the existing lane-changing models for automatic driving consider the LV as the main body to consider the lane-changing problem. There is no relevant research on the joint design mechanism for LV and RV vehicles to decide who should make an

avoidance move in case of conflict. The game-theoretic-based lane-changing conflict coordination model for automated vehicles proposed in this paper is more capable of judging who should avoid when LV and RV conflict

- The traditional lane change model only considers the kinematics characteristics of the vehicle. The lane change model in this paper combines game theory with kinematics so that the vehicle can complete the lane change decision under the premise of ensuring safety.

The rest of the paper is organized as follows. Section 2 reviews the existing studies relevant to the game among automated vehicles. Section 3 proposes conflict management models for the lane-changing of automated vehicles. Section 4 explores the effectiveness of the proposed model through simulations, and Section 5 concludes this paper and discusses future work.

## 2. Literature Review

With the rapid development of autonomous driving technology, there is more and more research on lane change strategies. Ding et al. [31] proposed a multi-vehicle co-operative lane change strategy for the driving environment on the roundabout. In this strategy, the driver's attack coefficient is defined according to the acceleration fluctuation of the vehicle. Then a cost function is proposed according to the driving state of the vehicle and the distance to the exit. Finally, the Stackelberg game is used to model the relationship between the LV and its surrounding vehicles to determine the lane change time. Wang et al. [32] designed a lane-changing feasibility criterion, considering the acceptable acceleration/deceleration of adjacent vehicles, and proposed a coordinated lane-change strategy based on model predictive control to realize centralized decision-making and the active coordination of lane-changing vehicles and the front and rear vehicles in the target lane. Falsone et al. [33] proposed a multi-vehicle coordination strategy for lane change coordination in multi-vehicle autonomous driving scenarios. The strategy considers the fleet as consisting of an infinite number of vehicles, which is implemented by solving a multiparameter optimization procedure.

At present, there are few studies on the game among automated vehicles. Some researchers used game theory to study the decision of lane-changing and car-following for the mandatory lane change (MLC) problem. Zhang et al. [34,35] proposed a game theory-based model for the lane-changing of automated vehicles considering the aggressiveness estimation of the RV, in which the proposed model first selected a target vehicle based on the Stackelberg equilibrium, estimated the target vehicle's aggressiveness based on the interaction between the SV and the target vehicle, and then completed the lane change process through model predictive control (MPC). Ali et al. [36] proposed a comprehensive mandatory lane change model, which was used to model the driver's merging behavior for traditional and connected environments. The model was calibrated using Next Generation Simulation (NGSIM) data and simulator data.

Some investigators have used game theory to solve the problem of discretionary lane-changing. Yu et al. [37] proposed a game theory-based lane-changing model for automated vehicles in the hybrid driving environment, in which the aggression was also introduced into payoff functions and the Stackelberg game was adopted. Meng et al. [38] proposed a dynamic decision method for the lane-changing of automated vehicles based on game theory. All possible trajectories of surrounding vehicles were calculated through an accessibility analysis to take into account their uncertainties for the lane-changing vehicles and were then used to calculate the payoff of game theory. Finally, the vehicle chose the strategies according to the Nash equilibrium. Kim et al. [39] proposed a vehicle motion decision model based on mixed-motivation game theory. The model was suitable for vehicles in the same lane or an adjacent lane in the freeway environment. The payoff matrix was defined by the willingness of the participating vehicles to each strategy and the safety of the strategic combination, and the final strategies were determined by the Nash equilibrium. Ding et al. [40] proposed a proactive-passive lane-changing framework based

on the Markov game. They regarded the problem as a Markov game between proactive and passive cars and applied deep reinforcement learning to solve the lane-changing problem.

Besides, a few studies have adopted game theory to establish models suitable for both mandatory lane-changing and discretionary lane changing. Wang et al. [41] proposed a prediction method of lane-changing and car-following control based on the unification of rolling time domain optimal control and dynamic game theory. The problem of lane-changing is described as a differential game, and the latest state information about controlling vehicles and surrounding vehicles was used for the decision and updated with a stable frequency. Talebpour et al. [42] proposed a lane-changing model based on game theory. The two game types were considered for modeling lane-changing behavior. The first one was a two-person non-zero-sum non-cooperative game under complete information in the presence of connected vehicle technology, and the other one was a two-person non-zero-sum non-cooperative game under incomplete information in its absence.

From the above review, it can be concluded that it is clear that the design concept of the decision algorithm of automatic driving lane-changing is to maximize individual payoff, and the research of lane-changing games also focuses on describing the dynamic game process between the LV and RV. Automated vehicles can still be completed through multiple games in the lane-changing process, but there are no rules for conflict management between the LV and RV. There are huge potential safety hazards in actual lane-changing.

In summary, different from previous studies, this paper models the conflict between lane-changing vehicles and vehicles in the target lane. In the case of conflict between two vehicles, the benefits of both vehicles are considered comprehensively instead of the benefits of vehicles changing lanes. Then the payoff under different strategy combinations is analyzed by game theory to obtain the final strategy.

## 3. Methodology

### 3.1. Model Framework

The framework of the proposed conflict management model for the lane-changing of automated vehicles is displayed in Figure 2. First, the LV judges whether there is a conflict between the LV and RV. Second, if a conflict exists, the payoffs of the two vehicles are calculated for the different strategy combinations, and the final chosen strategy for the two vehicles is obtained based on the proposed game theory model. Third, the LV and RV move forward according to the adopted strategies, directly adopting the proposed acceleration models.

### 3.2. Analysis of Lane-Changing Game

3.2.1. Potential Conflict Point

The game between the LV and RV in the lane-changing decision is generated when the two vehicles have spatial interest conflicts. As shown in Figure 3, the lane-changing trajectory of the LV is the red dotted line, while if the RV continues to move straight forward, its trajectory is the blue dotted line. The trajectory of the two vehicles will intersect at a certain point on the target lane. This point is defined as the potential conflict point in lane-changing, and the time difference of reaching the potential conflict point for the two vehicles is defined as the time difference to collision (TDTC) [43]. The small value of TDTC means that the safety of lane-changing is low. When the value of TDTC does not exceed a certain threshold, the two vehicles need to play a game to decide the final driving strategy. Thus, the potential conflict points between the LV and RV need to be calculated first.

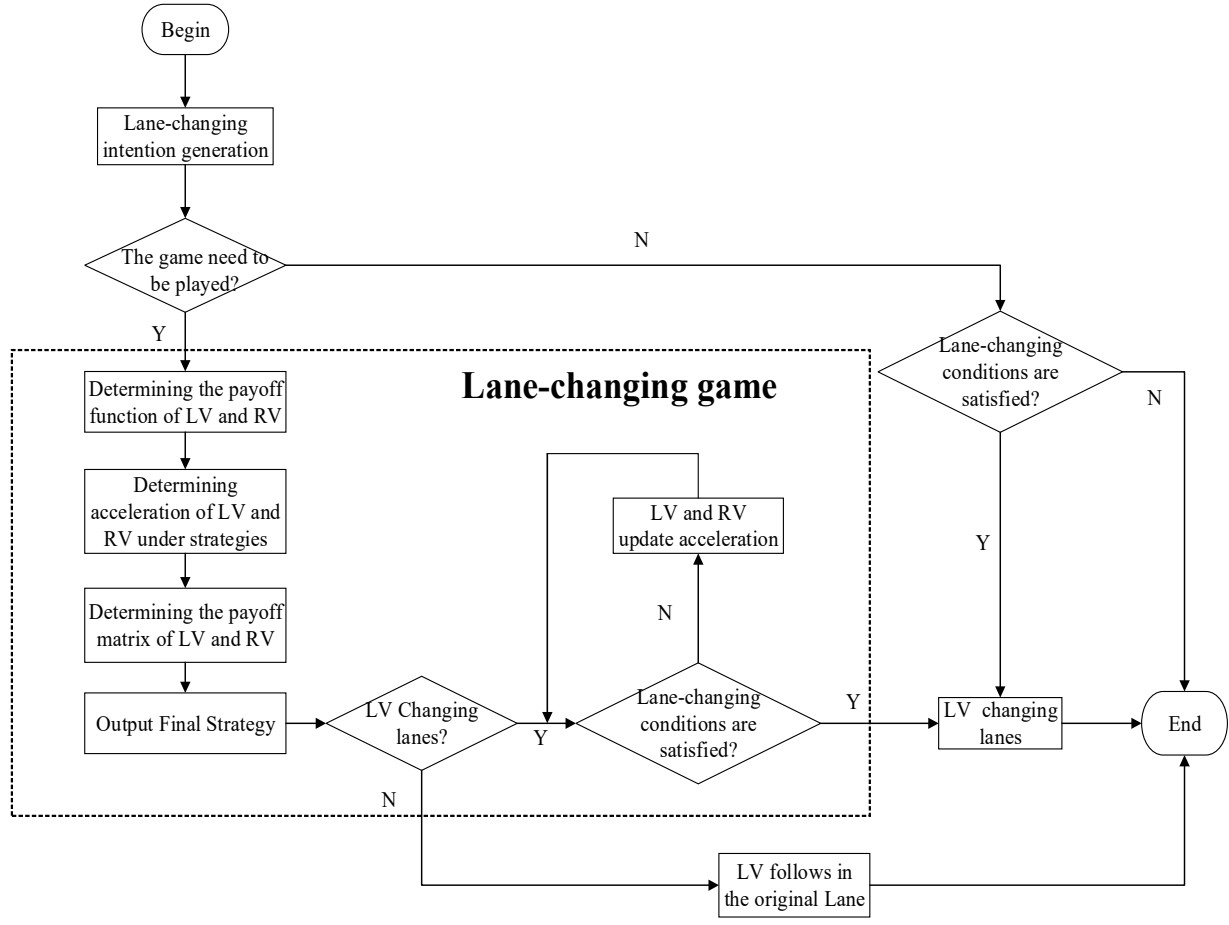

**Figure 2.** Framework of the proposed lane-changing conflict management model for automated vehicles.

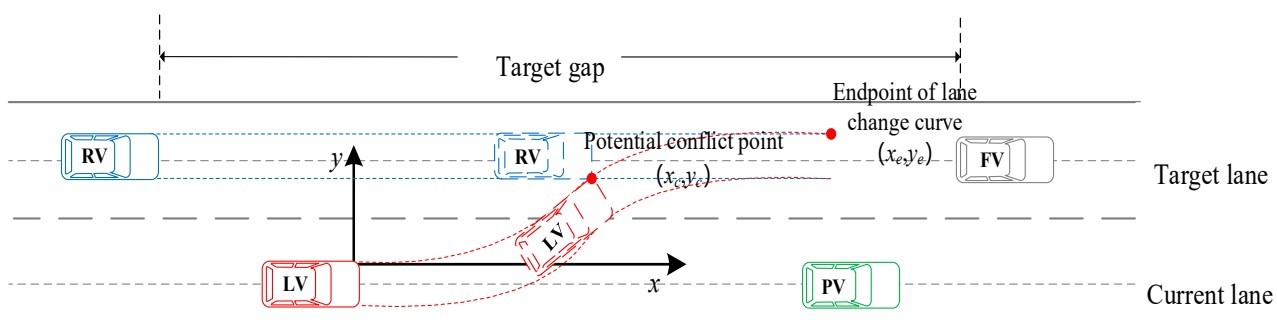

**Figure 3.** Potential conflict point in lane-changing.

Defining the coordinates of the potential conflict point as $(x_c, y_c)$ where $y_c$ is calculated as follows:

$$y_c = y_e - w_{car} \tag{1}$$

where, $y_e$ denotes the coordinate of the ending point of the lane-changing trajectory, and $w_{car}$ denotes the vehicle width.

In Equation (1), $y_e$ can be determined directly by the lane width, so $y_c$ is known. $x_c$ is determined from the lane-changing trajectory curve of the automated vehicle. The most commonly used lane-changing trajectory curve function is the polynomial curve, as follows:

$$y(x) = a_0 + a_1 x + a_2 x^2 + a_3 x^3 \tag{2}$$

where $x$ and $y$ denote the lateral and longitudinal positions of the left side of LV's front bumper, and $a_0$, $a_1$, $a_2$, $a_3$ are the parameters that need to be determined.

The parameters ($a_0$, $a_1$, $a_2$, $a_3$) are obtained using the method outlined in the reference [44], and the lane-changing trajectory equation was derived as follows:

$$y(x) = \frac{3y_e}{x_e^2}x^2 - \frac{2y_e}{x_e^3}x^3 \tag{3}$$

For a given $y_c$ value, $x_c$ of the potential conflict point can be calculated, and the position of the potential conflict point was finally obtained.

### 3.2.2. Game Conditions for Lane-Changing

Not all lane-changings need a game for the LV and RV. There are two conditions for the lane-changing game. The first is to keep a safe distance between the LV and PV, and the second is that the TDTC between the LV and RV does not exceed a certain threshold. The first condition is to ensure safety in the process of lane-changing. The second condition finds interest conflict between the LV and RV.

The Gipps' safe distance function [45] is used to calculate the safe distance between the LV and PV, as follows:

$$G_l = x_p(t) - x_l(t) - l_p = v_l(t)\tau_l + \frac{v_l(t)^2}{2b_l} - \frac{v_p(t)^2}{2b_p} \tag{4}$$

where $G_l$ denotes the safe distance between the LV and PV, $x_l(t)$ denotes the LV's position at the time $t$, $x_p(t)$ denotes the PV's position at the time $t$, $l_p$ denotes the vehicle length of the PV, $b_l$ denotes the maximum deceleration of the LV, $\tau_l$ denotes the reaction time of the LV, $v_l(t)$ denotes the velocity of the LV at the time $t$, and $v_p(t)$ denotes the velocity of the PV at the time $t$.

To calculate the TDTC between the LV and RV, it is necessary to calculate the time instances when the two vehicles arrive at the conflict point, respectively. The distance traveled by the LV from the starting point of the lane-changing trajectory to the conflict point is calculated as follows:

$$\begin{aligned} L_l &= \int_0^{x_c} ds = \int_0^{x_c} \sqrt{1 + y'^2}\,dx \\ &= \int_0^{x_c} \sqrt{1 + \left(2\frac{3y_e - 2x_e}{x_e^2}x + 3\frac{x_e - 2y_e}{x_e^3}x^2\right)^2}\,dx \end{aligned} \tag{5}$$

where $L_l$ denotes the distance from the starting point of the lane-changing trajectory to the conflict point for the LV.

The distance from the current position to the conflict point of the RV is calculated as follows:

$$L_r = x_c + d \tag{6}$$

where, $L_r$ denotes the distance from the current position to the conflict point for the RV, and $d$ denotes the longitudinal distance headway between the LV and RV.

Therefore, the TDTC is derived as follows:

$$\left| \frac{L_r}{v_r(t)} - \frac{L_l}{v_l(t)} \right| \le T_M \tag{7}$$

where, $T_M$ denotes the TDTC, and $v_r(t)$ and $v_l(t)$ denote the velocities of the RV and LV at the time $t$, respectively.

### 3.2.3. Strategies in a Lane-Changing Game

In the proposed lane-changing game model, the LV has two strategies of changing a lane and not changing a lane, and the RV has two strategies of avoiding the LV and not

avoiding the LV. When the LV chooses the strategy of changing a lane, it means that the LV can start moving from the current lane to the target lane. When the LV chooses the strategy of not changing a lane, the LV will stop a lane-changing temporarily and continue to search for the next lane-changing opportunity. When the RV chooses the strategy of avoiding, it will actively create a lane-changing condition for the LV. When the RV chooses the strategy of not avoiding, it will prevent the LV from changing a lane.

In addition, the lane-changing game belongs to the complete information static game. In the game, both the LV and RV are automated vehicles, and they have a clear understanding of the strategic space of each other and the payoff of each strategy combination. In the game process, the LV and RV will make a strategy decision simultaneously and play a one-shot game.

*3.3. Payoff Functions*

The LV and RV use the three payoff indicators to evaluate the strategies: the speed payoff, safety payoff, and comfort payoff, respectively.

3.3.1. Speed Payoff

1.    Speed payoff of the LV

The LV changes a lane to achieve the higher speed payoff, so the speed payoff needs to be defined and estimated by the LV for both of the two strategies of changing a lane and not changing a lane.

(1). Not changing a lane

When the LV chooses the strategy of not changing a lane, its speed will be limited by the PV, so the speed difference between the PV and LV can be used to represent the LV's speed payoff under the strategy of not changing a lane, as follows:

$$U_{vel}^{l,nc} = v_p(t) - v_l(t) \tag{8}$$

where $U_{vel}^{l,nc}$ denotes the LV's speed payoff for the strategy of not changing lanes, $v_p(t)$ denotes the speed of the PV at time $t$, and $v_l(t)$ denotes the speed of the LV at time $t$.

(2). Changing a lane

When the LV chooses the strategy of changing a lane, it will change the car-following target from the PV to the FV, so the LV's expected speed is the FV's speed. The speed difference between the FV and LV is the speed payoff of the LV for the strategy of changing a lane, as follows,

$$U_{vel}^{l,c} = v_f(t) - v_l(t) \tag{9}$$

where $U_{vel}^{l,c}$ denotes the LV's speed payoff for the strategy of changing a lane, and $v_f(t)$ denotes the FV's speed at time $t$.

2.    Speed payoff of the RV

In a lane-changing with interest conflict, if the RV chooses to avoid it, it has to decelerate, which will sacrifice its speed interest. Thus, it needs to calculate the speed payoff of the RV for the two strategies of avoiding and not avoiding.

(1). Avoiding

Assuming that the LV executes the lane-changing immediately, the RV will have an expected avoiding speed under the strategy of avoiding, which can just ensure the lane-changing safety of the LV. The expected avoiding speed of the RV can be derived by the following equation:

$$\frac{L_r}{v_r^a} - T_l(t) = T_M \tag{10}$$

where, $v_r^a$ denotes the expected avoiding speed of the RV, and $T_l(t)$ denotes the time of the LV traveling from the current position to the potential conflict point.

Hence, the expected avoiding speed of the RV can be calculated as follows:

$$v_r^a = \frac{L_r}{T_l(t) + T_M} \tag{11}$$

Assuming that the LV changes a lane at time $t$ with the speed $v_l(t)$ and acceleration $a_l(t)$, the distance between the LV and the potential conflict point $L_l$ can be derived as follows:

$$L_l = v_l(t)T_l(t) + \frac{1}{2}a_l(t)T_l^2(t) \tag{12}$$

Thus, the expression of $T_l(t)$ can be obtained as follows:

$$T_l(t) = \begin{cases} \sqrt{\frac{v_l^2(t)}{a_l^2(t)} + \frac{2L_l}{a_l(t)}} - \frac{v_l(t)}{a_l(t)} & a_l(t) > 0 \\ -\sqrt{\frac{v_l^2(t)}{a_l^2(t)} + \frac{2L_l}{a_l(t)}} - \frac{v_l(t)}{a_l(t)} & a_l(t) < 0 \\ \frac{L_l}{v_l(t)} & a_l(t) = 0 \end{cases} \tag{13}$$

The difference between the RV's expected avoidance speed and the RV's current speed is the RV's speed payoff for the avoiding strategy, as follows:

$$U_{vel}^{r,a} = v_r^a - v_r(t) \tag{14}$$

where $U_{vel}^{r,a}$ denotes the RV's speed payoff for the avoiding strategy.

(2). Not avoiding

If the RV adopts the strategy of not avoiding, the RV will continue to follow the FV, and the speed difference between the FV and RV is the speed payoff for the strategy of not avoiding, as follows:

$$U_{vel}^{r,na} = v_f(t) - v_r(t) \tag{15}$$

where $U_{vel}^{r,na}$ denotes the RV's speed payoff for the strategy of not avoiding.

### 3.3.2. Comfort Payoff

For automated vehicles, comfort is also a critical factor that needs to be considered. In the process of lane-changing, the variation of vehicle acceleration is the indicator of comfort. Thus, the vehicle's comfort payoff is described by the change in the acceleration of a vehicle between the two adjacent steps, that is:

$$U_{com}^l = \left| a_l(t) - a_l(t - \lambda) \right| \tag{16}$$

where $U_{com}^l$ denotes the LV's comfort payoff, and $a_l(t - \lambda)$ denotes the acceleration of the LV for the last step.

Similarly, the comfort payoff of the RV is described as follows:

$$U_{com}^r = \left| a_r(t) - a_r(t - \lambda) \right| \tag{17}$$

where $U_{com}^r$ denotes the RV's comfort payoff, and $a_r(t - \lambda)$ denotes the acceleration of the RV for the last step.

### 3.3.3. Safety Payoff

Safety is also very important in a lane-changing process. The safety payoff functions for the two strategies of changing a lane and not changing a lane for the LV are defined respectively.

1. Strategy of changing a lane

When the LV chooses the strategy of changing a lane, the safety payoff of the two conflicting vehicles can be calculated by TDTC. Assuming that the RV drives forward with the speed $v_r(t)$ and acceleration $a_r(t)$, if the travel time of the RV from the current position to the potential conflict point is $T_r(t)$, the distance of the RV from the current position to the potential conflict point can be calculated as follows:

$$L_r = v_r(t)T_r(t) + \frac{1}{2}a_r(t)T_r^2(t) \tag{18}$$

The expression of $T_r(t)$ can be deduced from Equation (19) as follows:

$$T_r(t) = \begin{cases} \sqrt{\frac{v_r^2(t)}{a_r^2(t)} + \frac{2L_r}{a_r(t)}} - \frac{v_r(t)}{a_r(t)} & a_r(t) > 0 \\ -\sqrt{\frac{v_r^2(t)}{a_r^2(t)} + \frac{2L_r}{a_r(t)}} - \frac{v_r(t)}{a_r(t)} & a_r(t) < 0 \\ \frac{L_r}{v_r(t)} & a_r(t) = 0 \end{cases} \tag{19}$$

The TDTC value between the LV and RV is as follows:

$$\Delta T = |T_r(t) - T_l(t)| \tag{20}$$

Considering the significance of safety to automated vehicles, the range of safety payoff is not limited to $[-1, 1]$ as that of speed and comfort payoffs. As shown in Figure 4, when the TDTC between the LV and RV is close to zero, that is, when the two vehicles almost reach the potential conflict point at the same time, the safety payoff tends to be negative infinity. In this case, with the increment of $\Delta T$, the safety payoff will increase as well, but the increasing rate will gradually decrease. When the value of $\Delta T$ continues to increase to the safety threshold $T_M$, the safety payoff reaches the maximum value with a return value of 0. After that, the safety payoff remains stable. This process can be described by a logarithmic function, and the expression of the safety payoff is as follows:

$$U_{safe}^{r,c} = U_{safe}^{l,c} = \begin{cases} -\infty & (\Delta T = 0) \\ \ln\left(\frac{\Delta T}{T_M}\right) & (0 < \Delta T < T_M) \\ 0 & (\Delta T \geq T_M) \end{cases} \tag{21}$$

where $U_{safe}^{r,c}$ and $U_{safe}^{l,c}$ denote the safety payoffs of the RV and LV under the lane-changing strategy of the LV, respectively.

2. Strategy of Not Changing Lane

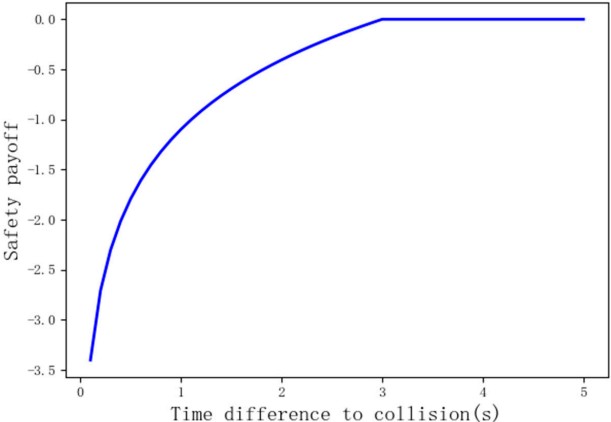

**Figure 4.** Relation between the safety payoff and TDTC.

When the LV chooses the strategy of not changing a lane, there is no conflict between the two vehicles. In this case, there is no safety risk for the RV and LV, and the safety payoff of both two vehicles is 0.

### 3.3.4. Total Payoff

Based on the functions of the speed, safety, and comfort payoffs of the LV and RV under each strategy, the total payoffs can be obtained. Different combinations of strategies for the LV and RV will produce different total payoff values. There are four strategy combinations, including $S_{11}$ = {Changing a lane, Avoiding}, $S_{12}$ = {Changing a lane, Not avoiding}, $S_{21}$ = {Not changing a lane, Avoiding}, and $S_{22}$ = { Not changing a lane, Not avoiding }. The payoff matrix of the four strategy combinations is shown in Table 1.

**Table 1.** Payoff Matrix.

|  |  | RV | |
| --- | --- | --- | --- |
|  |  | **Avoiding** | **Not Avoiding** |
| LV | Changing a lane | $(U_{11}^l, U_{11}^r)$ | $(U_{12}^l, U_{12}^r)$ |
|  | Not changing a lane | $(U_{21}^l, U_{22}^r)$ | $(U_{22}^l, U_{22}^r)$ |

Under the strategy combination $S_{11}$, the total payoffs of the LV and RV are as follows,

$$U_l^{11} = \alpha_1 f_{vel}(U_{vel}^{l,c}) + \beta_1 f_{com}(U_{com}^l) + \gamma_1 f_{safe}(U_{safe}^{l,c}) \tag{22}$$

$$U_r^{11} = \alpha_2 f_{vel}(U_{vel}^{r,a}) + \beta_2 f_{com}(U_{com}^r) + \gamma_2 f_{safe}(U_{safe}^{r,c}) \tag{23}$$

where $U_{11}^l$ and $U_{11}^r$ denote the total payoffs of the LV and RV under the strategy combination $S_{11}$, $f(*)$ denotes the normalized result of the payoff value, $\alpha_1$, $\beta_1$, and $\gamma_1$ denote the weight parameters for the speed, safety, and comfort payoffs of the LV, respectively, and $\alpha_2$, $\beta_2$, and $\gamma_2$ are the weight parameters for the speed, safety, and comfort payoffs of the RV, respectively.

Similarly, under the strategy combination $S_{12}$, the total payoffs of the LV and RV are as follows,

$$U_l^{12} = \alpha_1 f_{vel}\left(U_{vel}^{l,c}\right) + \beta_1 f_{com}\left(U_{com}^l\right) + \gamma_1 f\left(U_{safe}^{l,c}\right) \tag{24}$$

$$U_r^{12} = \alpha_2 f_{vel}(U_{vel}^{r,a}) + \beta_2 f_{com}(U_{com}^r) + \gamma_2 f_{safe}(U_{safe}^{r,c}) \tag{25}$$

where $U_{12}^l$ and $U_{12}^r$ denote the total payoffs of the LV and RV under the strategy combination $S_{12}$.

Under the strategy combination $S_{21}$, the total payoffs of the LV and RV are shown as follows:

$$U_l^{21} = \alpha_1 f_{vel}(U_{vel}^{l,nc}) + \beta_1 f_{com}(U_{com}^l) \tag{26}$$

$$U_r^{21} = \alpha_2 f_{vel}(U_{vel}^{r,a}) + \beta_2 f_{com}(U_{com}^r) \tag{27}$$

where $U_{21}^l$ and $U_{21}^r$ denote the total payoffs of the LV and RV under the strategy combination $S_{21}$.

Under the strategy combination $S_{22}$, the total payoffs of the LV and RV are as follows:

$$U_l^{22} = \alpha_1 f_{vel}(U_{vel}^{l,nc}) + \beta_1 f_{com}(U_{com}^l) \tag{28}$$

$$U_r^{22} = \alpha_2 f_{vel}(U_{vel}^{r,na}) + \beta_2 f_{com}(U_{com}^r) \tag{29}$$

where $U_{22}^l$ and $U_{22}^r$ denote the total payoffs of the LV and RV under strategy combination $S_{22}$.

### 3.4. Acceleration Model

3.4.1. Acceleration Model of the LV

1.  The LV chooses the strategy of changing a lane

In this case, the LV's acceleration is affected by the FV and RV. When the LV chooses the strategy of changing a lane, the LV will maintain safety distances with both the FV and RV. In this paper, a lane-changing acceleration model based on the desired time headway is introduced [46]. It is considered that the LV tends to choose an acceleration that can maintain the desired time headways with the FV and RV. The acceleration is determined by the difference between the current time headway and the desired time headway, as follows:

$$a_l^c(t) = k\left(h_f(t) - h_f^e(t)\right) + (1 - k)(h_r(t) - h_r^e(t)) \tag{30}$$

$$h_f^e(t) = \frac{a_1 + b_1 v_l(t) - c_1\left(v_f(t) - v_l(t)\right)}{v_l(t)} \tag{31}$$

$$h_r^e(t) = \frac{a_2 - b_2 v_l(t) + c_2(v_r(t) - v_l(t))}{v_r(t)} \tag{32}$$

where $a_l^c(t)$ denotes the acceleration of the LV under the strategy of changing a lane, $h_f(t)$ denotes the time headway between the LV and FV at time $t$, $h_r(t)$ denotes the time headway between the LV and RV at time $t$, $h_f^e(t)$ denotes the desired headway between the FV and LV at time $t$, $h_r^e(t)$ denotes the desired headway between the RV and LV at time $t$, $k$ denotes the contribution of the FV in determining the final acceleration of the LV, and $a_1$, $b_1$, $c_1$, $a_2$, $b_2$, and $c_2$ are the parameters.

2.  The LV chooses the strategy of not changing a lane

When the LV chooses the strategy of not changing a lane, the LV keeps following the PV on the original lane. In the car-following state, the LV and PV should keep a safe distance that can ensure that the LV and PV do not have a rear-end crash when the PV brakes emergently. Therefore, this paper introduces the Gipps' safe distance rule [33] to model the acceleration of the LV in this case. According to the Gipps' model, to avoid a collision with the PV, the safe following speed of the LV is as follows:

$$v_l^{nc}(t + \tau) = b_l \tau + \sqrt{b_l^2 \tau^2 - b_l\left[2(x_p(t) - l_{car} - x_l(t)) - v_l(t)\tau - v_p^2(t)/b_p\right]} \tag{33}$$

where $v_l^{nc}(t + \tau)$.enotes the safe speed of the LV following the PV, $b_p$ and $b_l$ denote the maximum braking accelerations of the PV and LV, $x_l(t)$ denotes the longitudinal position of the LV at time $t$, $x_p(t)$ denotes the longitudinal position of the PV at time $t$, and $\tau$ denotes the reaction time.

The acceleration of the LV under the strategy of not changing a lane is as follows:

$$a_l^{nc}(t) = \frac{v_l^{nc}(t + \tau) - v_l(t)}{\tau} \tag{34}$$

where $a_l^{nc}(t)$ denotes the acceleration of the LV under the strategy of not changing a lane.

3.4.2. Acceleration Model of the RV

1.  The RV chooses the strategy of avoiding

The acceleration of the RV under the strategy of avoiding is calculated by maximizing the total payoff of the RV. When the RV chooses the strategy of avoiding, there are two combinations of strategies, {change a lane, avoiding} and {not changing a lane, avoiding}.

The total payoffs of the two strategies are Equations (23) and (27), respectively. Thus, the optimal avoiding acceleration of the RV is obtained from the following equation:

$$\begin{cases} a_r^{11}(t) = \max\{U_{11}^r(a_r)\}, b_r \le a_r \le 0 \\ a_r^{21}(t) = \max\{U_{21}^r(a_r)\}, b_r \le a_r \le 0 \end{cases} \tag{35}$$

where, $a_r^{11}(t)$ and $a_r^{21}(t)$ denote the optimal avoiding accelerations of the RV under the strategies of $S_{11}$ and $S_{21}$, $b_r$ denotes the maximum deceleration of the RV, and $a_r^a$ denotes the avoiding acceleration of the RV.

2.  RV chooses the strategy of not avoiding

When the RV adopts the strategy of not avoiding, it will continue to follow the FV and keep a safe distance from the FV. Similarly, according to Gipps' model, the safe following speed of the RV is as follows:

$$v_r^{na}(t + \tau) = b_r\tau + \sqrt{b_r^2\tau^2 - b_r\left[2\left(x_f(t) - l_{car} - x_r(t)\right) - v_r(t)\tau - v_f^2(t)/b_f\right]} \tag{36}$$

where $v_r^{na}(t + \tau)$ denotes the safe speed of the RV in following the FV, $b_f$ denotes the maximum braking acceleration of the FV, $x_r(t)$ denotes the longitudinal position of the RV at time $t$, and $x_f(t)$ denotes the longitudinal position of the FV at time $t$.

The acceleration of the RV under the strategy of not avoiding is as follows:

$$a_r^{na}(t) = \frac{v_r^{na}(t + \tau) - v_l(t)}{\tau} \tag{37}$$

where, $a_r^{na}(t)$ denotes the acceleration that RV will choose under the strategy of not avoiding.

### 3.5. Final Strategy Decision

The flow chart of the final strategy selection is displayed in Figure 5.

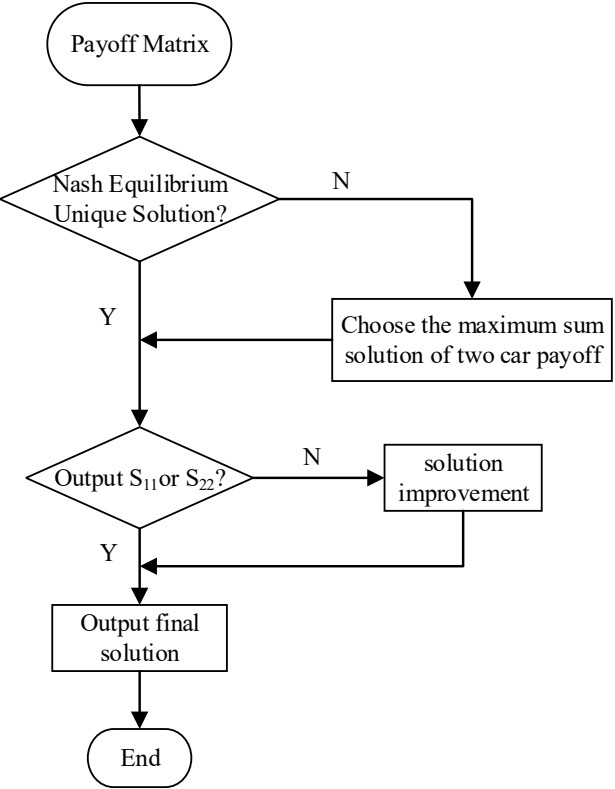

**Figure 5.** Final strategy selection flow chart.

First, the Nash equilibrium solution is solved by the payoff matrix. If there are multiple solutions in the Nash equilibrium, the solution that can maximize the total payoff of the two vehicles is selected. Second, it judges whether the output solution is {changing a lane, avoiding} or {not changing a lane, not avoiding}. If it is, the solution is the final strategy; if not, the strategy is improved to ensure that one of the two vehicles in the final solution makes a compromise.

Table 2 displays the payoff matrix of an example of a lane-changing game for an automated vehicle. From this table, we can observe that there are two Nash equilibrium solutions: {changing a lane, avoiding} and {not changing a lane, not avoiding}. Therefore, it is necessary to further determine the optimal strategy.

**Table 2.** Nash Equilibrium.

|  |  | RV | |
|---|---|---|---|
|  |  | **Avoiding** | **Not Avoiding** |
| LV | Changing a lane | $(0.10, -0.54)$ | $(-0.41, -0.60)$ |
|  | Not changing a lane | $(-0.10, -0.30)$ | $(-0.10, -0.04)$ |

When there are multiple Nash equilibrium solutions, the strategy with the maximum payoff sum of the LV and RV is selected. In the example in Table 2, the sum of the two vehicles' payoffs for the strategy combination {Changing a lane, Avoiding} is $-0.44$, and the sum of the two vehicles' payoffs for the strategy combination {Not changing a lane, Not avoiding} is $-0.14$, so the LV and RV will choose the strategy combination {Not changing a lane, Not avoiding}.

However, it should be noted that when the final solution is {Changing a lane, Not avoiding} or {Not changing a lane, Avoiding}, the two vehicles need to re-play the game, which will affect the driving efficiency of the automated vehicle. This situation should be avoided in the design of a lane-changing algorithm for automated vehicles. Thus, when the final solution is {Changing a lane, Not avoiding}, we need to improve the final solution to {Changing a lane, Avoiding} or {Not changing a lane, Not avoiding}. Assuming that the maximum acceptable reduction of the RV's payoff is $\theta$, when $U_r^{12} - U_r^{11} \geq \theta$, the solution {Changing a lane, Not avoiding} is improved to {Changing a lane, Avoiding}; when $U_r^{12} - U_r^{11} < \theta$, the solution {Changing a lane, Not avoiding} is improved to {Not changing a lane, Not avoiding}; when the final solution is {Not change a lane, Avoiding}, the final solution can be directly improved to {Not changing a lane, Not avoiding}.

## 4. Model Evaluations

### 4.1. Key Parameter Determination

The weight parameters for speed, safety, and comfort are three key parameters in this paper. To determine these parameters, we use a driver simulator and SCANeR studio to obtain the suitable values. The driver simulator can simulate a realistic vehicle dynamics system in several traffic environments. In the SCANeR studio, four vehicles (PV, FV, RV, and LV) are set on a road with two lanes, and SV is an automated vehicle, LV is a human-driven vehicle, as shown in Figure 6. The PV and FV are controlled by the model based on the desired speed, the SV is controlled by the proposed model, and the LV is driven by 20 volunteers using a driving simulator. After many tests, we chose the weights of speed payoff, safety payoff, and comfort payoff as $\alpha = 0.3$, $\beta = 0.5$, and $\gamma = 0.2$.

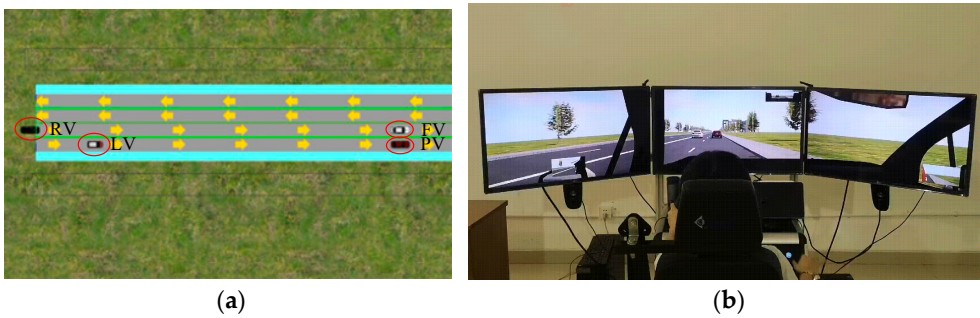

(**a**)                                                    (**b**)

**Figure 6.** Model parameter generation by SCANeR studio. (**a**) Initial positions of vehicles. (**b**) Experiments on the driver simulator.

### 4.2. Strategy Choice Evaluation

In the simulation evaluations, the simulation setup is displayed in Table 3, based on which the strategy choices of the LV and RV are simulated and analyzed.

**Table 3.** Initial State of Vehicles.

| Initial Position of Vehicle (m) | LV Position<br>90 | RV Position<br>(0, 90) | PV Position<br>180 | FV Position<br>180 |
|---|---|---|---|---|
| Initial Vehicle Speed (km/h) | LV Speed<br>90 | RV Speed<br>110 | PV Speed<br>90 | FV Speed<br>120 |

The simulation results exhibit that there are two Nash equilibrium solutions in the model, which are $S_{11}$ = {Changing a lane, Avoiding} and $S_{22}$ = {Not changing a lane, Not avoiding}. This indicates that the model can describe the conflicted relationship between the LV and RV in the lane-changing process.

According to the strategy selection rule in Section 3.5, it needs to further compare the payoff sum of the two vehicles under the two strategies to determine the final strategy. Figure 7 illustrates the relationship between the change in the RV's initial position and the payoff sum of the two vehicles. When the RV is initially located in the range of [0, 40 m], the payoff sum of the two vehicles under the strategy combination $S_{11}$ is the largest, so the final solution is $S_{11}$. The reason for this phenomenon is that the distance between the LV and RV is larger. It is easier for the LV to change a lane, and for the RV it is relatively difficult to prevent the LV from changing a lane. As the initial position of the RV increases, the distance between the LV and RV decreases, and the condition is more and more rigorous for lane-changing. Thus, when the initial position of the RV is in the range of [41 m, 90 m], the payoff sum under strategy $S_{22}$ exceeds that of strategy $S_{11}$, so the final solution changes from $S_{11}$ to $S_{22}$. The LV abandons the lane-changing attempt, and the two vehicles will continue to stay on the original lanes. The above simulation results indicate that the lane-changing rules designed in this paper are reasonable and can solve the lane-changing conflict between the LV and RV.

### 4.3. Safety Evaluation

To show the applicability of the proposed model, we compared TDTC before and after applying the proposed model. In the simulations, we change the speeds of the PV, FV, RV, and LV in the range of [30 km/h, 120 km/h] and run it 100 times for different speeds of the PV, FV, RV, and LV. The counted TDTC is displayed in Figure 8. From the figure, we can observe that, after applying the proposed model, the average value and variance of TDTC are significantly reduced. Thus, the proposed conflict management model can make the lane-changing of automated vehicles safer.

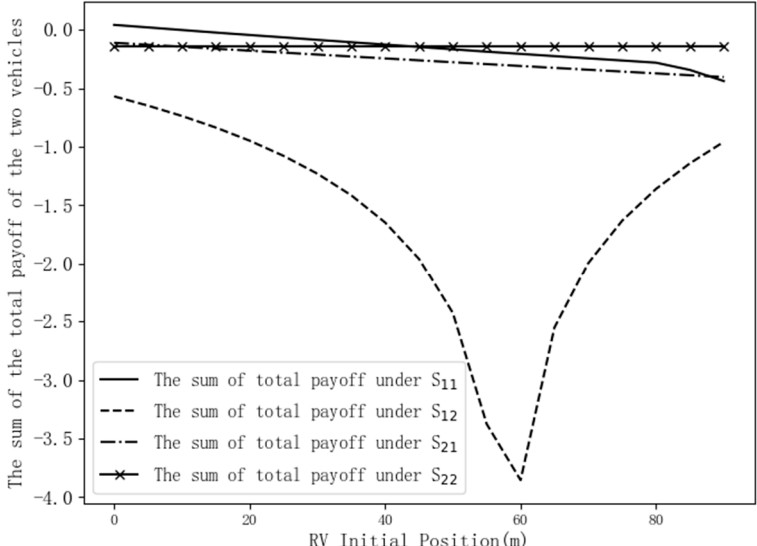

**Figure 7.** Relationship between the RV's initial location and the sum of two vehicles' payoffs.

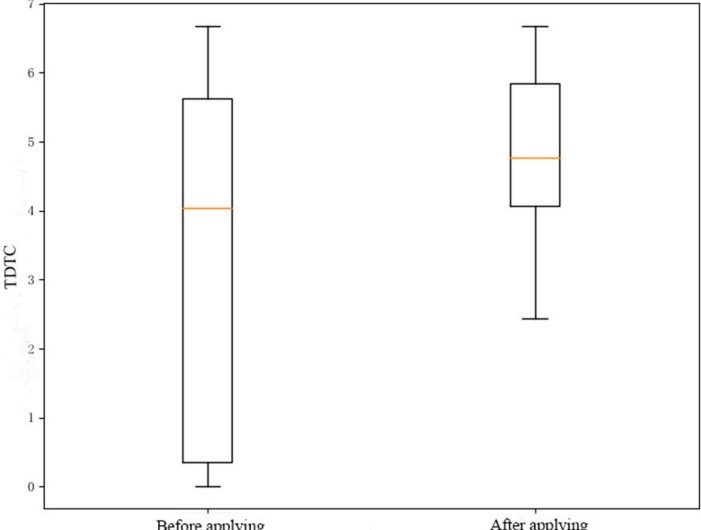

**Figure 8.** Box plot of conflict time difference at lane change time.

Figure 9 displays the changes in the safety and TDTC before and after applying the proposed model for the lane-changing of automated vehicles. Since there is no conflict relationship between the two vehicles when the LV chooses the strategy of not changing a lane, there is not a TDTC curve in the range of [40 m, 90 m] in Figure 9. From the figure, it can be observed that when the proposed model is not applied, with the increment of the initial position of the RV, the TDTC between the two vehicles decreases to zero first and then increases. In this case, the corresponding safety payoffs also show the same variation trend, and all of them are negative, which indicates that the two vehicles have greater potential safety risks. After applying the proposed model, when the initial position of the RV changes within the range of [0, 40 m], the LV chooses the strategy of changing a lane. The TDTC is near the safety threshold ($T_M$ = 3 s), so the maximum safety payoff is 0.

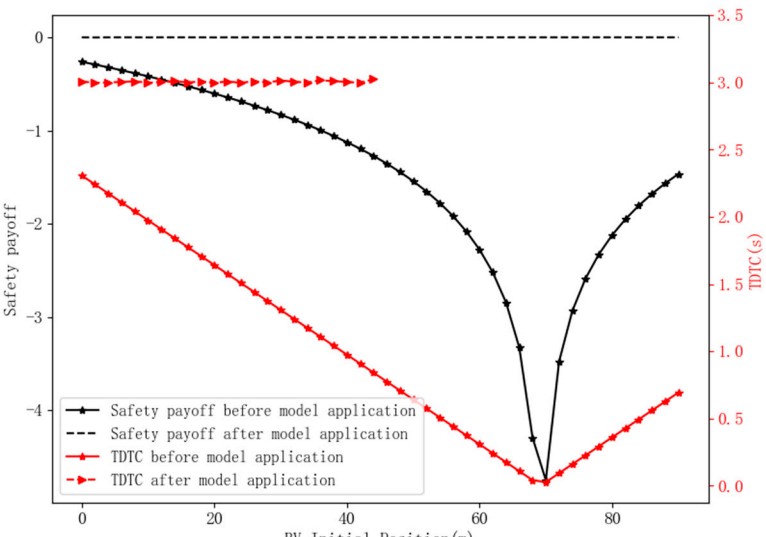

**Figure 9.** Safety analysis for the proposed model.

When the initial position of the RV is more than 40 m, the LV chooses the strategy of not changing a lane, the two vehicles continue following their preceding vehicles on the current lanes without lane-changing conflicts and potential safety risks. The above simulation results indicate that the safety of the two vehicles can be guaranteed in different scenarios when the proposed model is applied.

Furthermore, Figure 10 displays an example to illustrate the safety improvement when the proposed model is applied. Figure 10a,b respectively exhibit the lane-changing processes for the two cases of applying and not applying the proposed model. Figure 10a shows that the RV does not avoid the LV, so the TDTC between the two vehicles is very small. When the LV reaches the potential conflict point, the RV's following distance is far less than the safe distance, and there was a large potential safety risk between the two vehicles. In Figure 10b, when vehicle LV changes lanes, RV chooses an avoidance strategy, RV and LV maintain a safe distance through the acceleration model proposed in this paper, so that the safeties of both vehicles can be guaranteed from the beginning to the end of the lane-changing.

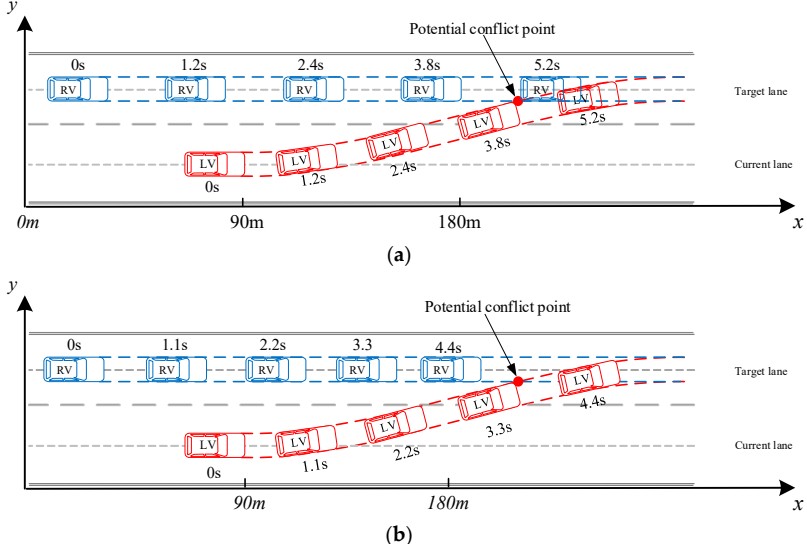

**Figure 10.** Lane-changing processes before and after the proposed model is applied. (**a**) Before the model is applied. (**b**) After the model is applied.

## 5. Conclusions and Future Work

With the rapid development of automatic driving technology, the research on the algorithm of automatic driving lane change systems still needs to be further improved. Most of the existing lane-changing models for automatic driving consider the LV as the main body to consider in the lane-changing problem. There is no relevant research on the joint design mechanism for LV and RV vehicles to decide who should make an avoidance move in the event of a conflict.

In this paper, after the lane-changing vehicle determines the target gap, it considers whether there are conflicts between the lane-changing vehicle and the vehicle behind the target lane and who should avoid it, analyzes the benefits that should be considered during the lane-changing process, and establishes the relationship between the two vehicles based on game theory and kinematics. The lane-changing model is used to obtain the final strategy of the two vehicles and the acceleration that should be selected under the strategy combination. Finally, this paper uses a simulation to analyze the influence and reason of the initial position and speed changes of LV and RV on the revenue of the two vehicles and the final solution. The main conclusions are drawn as follows:

(1) The model evaluation results indicate that the proposed model can reasonably manage the lane-changing conflict of automated vehicles. The designed mechanism in this paper not only can ensure the fairness of the LV and RV, but also can enhance the lane-changing efficiency and safety of automated vehicles.

(2) The TDTC is not only related to the distance between the two vehicles, but also to the speed and acceleration of the two vehicles, which can more truly and accurately reflect the safety payoffs.

(3) The logarithmic function relationship between safety payoffs and TDTC can magnify safety risks in lane-changing. When the safety risks are large, the safety payoff can even determine the trend of the total payoffs, which enhance the safety of automated vehicles in lane-changing.

In the real world, when the vehicle changes lanes, it may conflict with the RV, and in this case, the results obtained by the model analysis in this paper are used to perform operations, which avoids the problem that accidents may occur during the mutual game between LV and RV and can improve the safety of LV and RV in the process of lane-changing.

This paper also has some limitations. First, this paper only managed the lane-changing conflict between the two automated vehicles, and it will be expanded to deal with the conflict problem of multiple vehicles. Second, the impact of lane-changing on the overall traffic flow may also be incorporated into the game payoff function in future.

**Author Contributions:** Conceptualization, B.Z.; methodology, L.Z.; formal analysis, L.Z.; investigation, D.Y.; data curation, Z.C.; writing-original draft, X.Y. All authors have read and agreed to the published version of the manuscript.

**Funding:** This work was supported by Natural Science Foundation of Sichuan Province (Grant No. 2023NSFSC0905), the National Natural Science Foundation of China (Grant No. 52172333), Key Research and Development Project of Chengdu City (Grant No. 2022YF0500376SN).

**Institutional Review Board Statement:** Not applicable.

**Informed Consent Statement:** Not applicable.

**Data Availability Statement:** Authors elects to not share data.

**Conflicts of Interest:** The authors declare no conflict of interest.

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
