# Peer review of "A Model to Manage the Lane-Changing Conflict for Automated Vehicles Based on Game Theory"

_sustainability, doi:10.3390/su15043063_

Round 1
Reviewer 1 Report
1. There are many studies focusing on the problem of gaming lane change for autonomous vehicles, and the novelty and main contributions of the model should be highlighted in the introduction section.
2.P2 L73, The paper mentions that“By applying our method, automated vehicles encountering conflicts in the process of lane-changing do not need to really play games, but drive directly according to the results derived from the proposed game theory model in this paper.”The model in this paper is based on the assumption of complete information, and It is best to rephrase or replace this sentence considering the complexity of the actual scenario.
3.P14, Figure 6 and Figure 7 have a large size difference, it is better to carry out proportional unification.
Reviewer 2 Report
This paper has developed a new model for safe lane-changing maneuvers in an AV environment. Overall, the paper is well-written and well-organized. But the reviewer has a few comments:
- The authors should not use abbreviations in abstracts since they have not been defined before, e.g., LV and RV.
- The authors expanded the models very comprehensively. However, the simulation has not been covered well. How was the simulation developed? Which software was used?
- The conclusion section is very short and brief. It should be organized in a way that conveys the problem, methodology, and results of the paper. But it mainly has focused on the results. Please expand this section more.
Reviewer 3 Report
Please see the attachment.

Reviewer 4 Report
The proposed model has some merits for publication, however, there are certain limitations in the proposed model, including the following:
Abstract:
Please describe LV and RV
Introduction
Please provide references for the following statements:
-
Line 46, suggested it is difficult for automated vehicles to deal with such a situation without an elaborate conflict management design. Therefore, this paper attempts to design a lane-changing conflict management strategy to help automated vehicles change lanes safely when there is a conflict between the LV and RV.
-
Line 55. The third method is that the RV decelerates to produce a safe space for the LV.
-
Line 56-65.
-
Line 68-76.
Literature review
The authors have reviewed the existing literature on the lane change model using game theory. However, the authors failed to indicate the research gaps that need to be filled and how the proposed model can fill that research gaps. Most importantly, it is strongly recommended to present how the proposed method differs from the existing studies.
Methodology
The proposed method adopted the Gipp lane change model
Most importantly, it is strongly recommended to present how the proposed method differs from the existing studies.
Look ahead
The proposed method only considers the speed and the position of LV and RV. However, lane change is a very complex process; thus this proposed method have a very limited application. I would strongly suggest authors include the review about look ahead lane change strategy.
In equations 22 and 23, the authors introduced the weight parameters for speed, safety and comfort to determine the total payoff. However, the methodology section does not provide enough information on how to generate these values and methods to validate them.
Please state the process to validate all parameters and state all assumptions.
The proposed model was tested using very limited data. Hence it is strongly recommended to test the proposed model to a wide range of conditions to show its applicability of the proposed model.
Given these limitations, unfortunately, it appears that the paper presents a relatively straightforward simulation study of connected vehicles on ramps. I do not see any scientific innovation or rigorous analysis.
Round 2
Reviewer 3 Report
(1) Please consider to strengthen the relevance to the journal by showing that related studies have been published in Sustainability before (2-3 papers).
(2) Please check the language one more time that few typos are still present.
(3) Please consider cite the following publications https://doi.org/10.1080/00207543.2022.2121870 Risk-averse two-stage stochastic minimum cost consensus models with asymmetric adjustment cost. Group Decis. Negot. 2022, 31, 261–291
(4) In the conclusion section, please strengthen the possible application in the real-world problems.
Reviewer 4 Report
The paper can be accepted in the current format.
Author Response
Thank you for your review.